# Increase of Prevalence of Obesity and Metabolic Syndrome in Children and Adolescents in Korea during the COVID-19 Pandemic: A Cross-Sectional Study Using the KNHANES

**DOI:** 10.3390/children10071105

**Published:** 2023-06-23

**Authors:** Jung Eun Choi, Hye Ah Lee, Sung Won Park, Jung Won Lee, Ji Hyen Lee, Hyesook Park, Hae Soon Kim

**Affiliations:** 1Department of Pediatrics, College of Medicine, Ewha Womans University, Seoul 07804, Republic of Korea; cjeped@ewha.ac.kr (J.E.C.); jwped@ewha.ac.kr (J.W.L.); major1106@naver.com (J.H.L.); 2Clinical Trial Center, Ewha Womans University Mokdong Hospital, Seoul 07985, Republic of Korea; khyeah@ewha.ac.kr; 3Department of Pediatrics, MizMedi Hospital, Seoul 07639, Republic of Korea; swped@naver.com; 4Graduate Program in System Health Science and Engineering, Department of Preventive Medicine, College of Medicine, Ewha Womans University, Seoul 07804, Republic of Korea; hpark@ewha.ac.kr

**Keywords:** COVID-19, obesity, metabolic syndrome, KNHANES

## Abstract

(1) Background: The aim of this study was to evaluate the prevalence of obesity and metabolic syndrome since the COVID-19 pandemic outbreak utilizing representative data on youth aged 2–18 years from the Korean National Health and Nutrition Examination Surveys (KNHANES) conducted in 2019–2020. (2) Methods: The survey consists of three parts: health interviews, health examinations, and nutrition surveys. From the 2019 and 2020 surveys, 1371 (2–9 years = 702 and 10–18 years = 669) and 1124 (2–9 years = 543 and 10–18 years = 581) individuals were included in the analysis. (3) Results: The mean body mass index (BMI) increased significantly among youth aged 2–9 years from 16.53 kg/m^2^ in 2019 to 17.1 kg/m^2^ in 2020 (*p* < 0.01). In youth aged 10–18 years, the BMI was found to increase slightly from 21.25 kg/m^2^ in 2019 to 21.41 kg/m^2^ in 2020 (*p* = 0.64). The increasing prevalence of extreme obesity was significant in girls, especially those aged 2–9 years (*p* < 0.01). However, extreme obesity had increased in 10–18-year-old boys (*p* = 0.08). The overall prevalence of metabolic syndrome in adolescents increased from 3.79% to 7.79% during the COVID-19 pandemic (*p* = 0.01). (4) Conclusions: We observed that the prevalence of obesity and metabolic syndrome among children and adolescents has increased after the COVID-19 outbreak. This is believed to be associated with an increase in the rate of early comorbidities in adulthood. The prevention of the progression of pediatric obesity has recently become an urgent public health concern in Korea.

## 1. Introduction

In December 2019, the coronavirus infection (COVID-19) was first confirmed in Wuhan, China and spread rapidly worldwide. On 11 March 2020, the World Health Organization declared the COVID-19 infection a pandemic. Since the COVID-19 outbreak, there have been many lifestyle changes [1]. The government mandated several policies to prevent contact among people, resulting in families spending more time at home. Online schooling started on 9 April 2020 for the first time in Korea due to the need for social distancing. Private gatherings were limited, and the business hours of malls and restaurants were reduced, while outdoor activities were also restricted [2]. Isolation plays a crucial role in preventing the transmission of the virus and facilitating recovery for individuals who have contracted COVID-19 [3].

As children and adolescents spent more time at home in online classes, their physical activities and eating behaviors also changed to avoid outdoor activities. This led to an increase in the consumption of delivery food, fast food, snacks, and high-calorie foods, as well as an increase in the time spent eating alone [4]. Furthermore, children of low socioeconomic status often lack access to a healthy diet, including fruits, vegetables, and whole-grain foods that are typically provided during the school years. Because vitamin D plays a crucial role in bone formation and mineralization, its deficiency in children can lead to the development of various diseases [5]. A balanced and healthy diet can supply adequate amounts of water, antioxidants, and fiber, all of which play a role in regulating risk factors associated with COVID-19 complications such as diabetes, hypertension, and weight gain [6]. These factors lead to imbalance in nutritional intake, reduced physical activity, and increased depression and stress called the corona blues, resulting in an accelerated weight gain in a short time. During the COVID-19 outbreak, 56.7% of school-aged children in Korea experienced weight gain [7].

Obesity is a major risk factor for metabolic syndrome (MetS), thereby increasing the non-alcoholic fatty liver disease (NAFLD) and cardiovascular disease, which can persist into adulthood [8]. The percentage of children affected by obesity has increased from 5% from 1963 to 1965 to 19% from 2017 to 2018 in the US. Between 2019 and 2021, the COVID-19 pandemic significantly affected overweight and obesity in children and adolescents [9]. The prevalence of MetS has increased in the last decade according to the increasing trend of childhood and adolescent obesity [10,11]. The COVID-19 pandemic has been a trigger to accelerate the increase [4,8]. Thus, efforts should be focused on preventing and reducing obesity in children and adolescents as a public health initiative [12].

Subjective surveys on lifestyle changes and weight gain after the COVID-19 outbreak have been conducted in Korea. However, few studies have been conducted on the prevalence of childhood obesity and MetS using objective body mass index (BMI) data and MetS components. Kim et al. reported that the prevalence of overweight and obesity among children aged 2 to 18 years in the KNHANES increased from 17.9% in 2001 to 18.6% in 2017 [11]. However, until now, there have been single-center studies on the prevalence of obesity and MetS components before and after the COVID-19 outbreak, but there has been no research yet on large-scale representative data for Korean children and adolescents [13,14].

Therefore, we aimed to investigate the prevalence of obesity and MetS before and during the COVID-19 pandemic outbreak employing representative data on youth aged 2–18 years from Korean National Health and Nutrition Examination Surveys (KNHANES) conducted from 2019–2020.

## 2. Materials and Methods

### 2.1. Data Sources and Study Subjects

The data for this study were obtained from the 2019 and 2020 KNHANES conducted by the Korea Centers for Disease Control and Prevention (KCDC). These are cross-sectional survey data representative of the country. The KNHANES targets non-institutionalized Korean citizens residing in Korea, and sampling follows a multistage, clustered probability design. The survey consists of three parts: health interviews, health examinations, and nutrition surveys. Detailed information on KNHANES has been published elsewhere [15]. In the 2019 and 2020 surveys, 1449 and 1174 individuals aged 2–18 years participated in the survey, respectively, of whom those who participated in health interviews (questionnaires on household income, parental education, physical activity, stress status, smoking, etc.) and health examinations (body measurements, blood pressure, laboratory tests, etc.) were included in this study. Thus, from the 2019 and 2020 surveys, 1371 (2–9 years = 702 and 10–18 years = 669) and 1124 (2–9 years = 543 and 10–18 years = 581) individuals aged 2–18 years were included in the analysis, respectively (Figure 1).

### 2.2. Definition of Study Variables

Body mass index was calculated by dividing the measured body weight (kg) in kilograms by height in meters squared (m^2^). Based on sex- and age-specific percentiles of BMI, the BMI status was defined as normal (<85th percentile), overweight (85–94th percentile), obesity (≥95th percentile), and extreme obesity (≥99th percentile). Classification was based on the 2017 Korean Children and Adolescent National Growth Charts [16].

Physical activity, smoking, stress, and sleep time were the lifestyle behaviors considered in this study. Subjects who performed physical activity for more than 60 min daily in the past 7 days were defined as the regular physical-activity group [17]. Subjects who performed muscle training for more than 2 days a week were defined as the regular-strength training group [18]. The average daily sedentary time was also determined. For sleep time, the average time of weekdays was used, excluding that of weekends. Adolescent smoking was defined as any use within the past 30 days [19]. Subjects who responded that they felt a lot of stress in their daily lives were defined as the recognizing-stress group. All these parameters were surveyed in subjects 12 years of age and older. Consumption of soft drinks and sports drinks was investigated in the 2019–2020 survey. The frequency of intake per week was calculated by considering the frequency and the average amount of intake per serving and was categorized according to whether these drinks were consumed more than 3 times a week [20]. We also considered dietary supplement use, which surveyed whether dietary supplement intake had occurred in the past 2 weeks.

For the daily nutritional intake of macronutrients, including total energy intake, we employed data from 24-h recall surveys. Excessive caloric intake for sex and age was defined based on the 2015 Dietary Reference Intakes for Koreans [21].

### 2.3. Definition of Metabolic Syndrome

Serological tests were performed on subjects aged 10 years and older. Therefore, in this study, MetS was defined for children aged 10–18 years when three or more of the following criteria were met: waist circumference (WC) ≥ 90th percentile for sex and age, systolic blood pressure (BP) or diastolic BP ≥ 90th percentile for sex, age, and height, fasting blood glucose (FBG) ≥ 100 mg/dL, triglycerides (TG) ≥ 110 mg/dL, and high-density lipoprotein-cholesterol (HDL-c) < 40 mg/dL. Elevated WC and elevated BP were defined based on the 2007 Korean Children and Adolescent National Growth Charts [22]. BP was measured three times in a sitting position, and the average of the secondary and tertiary measurements was used. Serological indicators were evaluated according to the above-mentioned criteria only when the fasting time was 8 h or more. Furthermore, a continuous metabolic syndrome (cMetS) score was calculated based on the metabolic components mentioned above employing the z-score method. The five key metabolic syndrome variables are used in the calculation of the score [23]. A higher score indicates a higher risk for metabolic syndrome, while a lower score indicates a normal metabolic status. To detect changes, the cMetS values of the study subjects in 2020 were calculated utilizing the mean and standard deviation information on the metabolic components of the subjects in 2019. A high cMetS score indicates an unfavorable metabolic health status [24].

### 2.4. Statistical Analysis

Data analyses were conducted employing SAS version 9.4 (SAS Institute, Cary, NC, USA). All statistical analyses were performed based on the multistage complex sampling survey design of the KNHANES. Based on the raw data usage guidelines provided by KCDC, ‘wt_itvex’ was basically used as a weight variable, and ‘wt_tot’ was used to calculate estimates for eating behavior and dietary intake. Summary statistics are presented as weighted means with 95% confidence intervals (CIs) for numeric variables and unweighted frequencies with weighted percentages for nominal variables. A general linear model for numeric variables and Rao–Scott Chi-square tests for nominal variables were used to analyze differences in basic characteristics, lifestyle behaviors, BMI status, and MetS components before and after the COVID-19 pandemic outbreak. The difference in the prevalence of MetS components before and after the outbreak was evaluated by sex. Additionally, differences in dietary nutrition and lifestyle factors were evaluated separately according to MetS before and after the COVID-19 outbreak. Additionally, we estimated crude and adjusted odds ratios for potential risk factors associated with metabolic syndrome with *p* < 0.1 in univariate analysis. Statistical significance was set at *p* < 0.05, using a two-tailed test.

## 3. Results

### 3.1. Comparison of Changes in Basic Characteristics of Study Subjects before and after the COVID-19 Pandemic Outbreak

Table 1 shows the differences in the characteristics of the study subjects in 2019 and 2020 before and after the COVID-19 outbreak. No differences in demographic characteristics were observed between 2019 and 2020. The mean BMI increased significantly among youth aged 2–9 years from 16.53 kg/m^2^ in 2019 to 17.1 kg/m^2^ in 2020 (*p* < 0.01). In youth aged 10–18 years, the BMI was found to increase slightly from 21.25 kg/m^2^ in 2019 to 21.41 kg/m^2^ in 2020 (*p* = 0.64).

There was no significant difference in physical activity, stress, and sleep time as lifestyle behaviors before and after the COVID-19 outbreak. In terms of nutritional factors, the overall energy intake at the age of 10–18 years decreased significantly from 2016.15 kcal in 2019 to 1910.08 kcal in 2020. However, no significant difference was observed in the intake of other macronutrients.

Among children aged 2–9 years, the prevalence of obesity increased in 2020 compared to 2019, with a notable increase observed among girls (*p* < 0.01). Among adolescents aged 10–18 years, there was no overall increase in the prevalence of obesity. However, an increase in the prevalence of severe obesity was observed, specifically among boys aged 10–18 years. (*p* < 0.08) (Figure 2).

### 3.2. Comparison of Changes in MetS Components before and after the COVID-19 Outbreak

Among the MetS components, diastolic BP, TGs, and HDL-c showed significant changes in youth aged 10–18 years. Diastolic BP increased from 66.78 mmHg to 68.56 mmHg (*p* = 0.02), while the TG value increased from 75.52 mg/dL to 83.97 mg/dL (*p* < 0.01). The mean HDL-c decreased from 52.62 mg/dL to 51.42 mg/dL (*p* = 0.09) (Table 2). In this regard, it was found that the change in the prevalence of MetS significantly increased from 3.79% to 7.79% since the COVID-19 outbreak (*p* = 0.01). There was no significant difference in the prevalence of MetS according to the BMI categories (Figure 3), but a higher obesity level and higher prevalence of MetS was consistently observed since the COVID-19 pandemic outbreak. The WC of 6–9-year-old children increased significantly from 57.67 cm in 2019 to 59.04 cm in 2020 (*p* = 0.05). The proportion of abdominal obesity (WC ≥ 90th percentile) of 6–9-year-old children also increased from 12.08% in 2019 to 17.23% in 2020, showing borderline significance (*p* = 0.09).

### 3.3. Comparison of Changes in Metabolic Syndrome Components before and after the COVID-19 Pandemic Outbreak by Sex

Table 3 shows a comparison of the MetS components before and after the COVID-19 pandemic outbreak according to sex. The WC of boys aged 10–18 years increased from 75.03 cm in 2019 to 76.27 cm in 2020 (*p* = 0.22), and the proportion of abdominal obesity also increased significantly from 12.52% to 19.83% (*p* = 0.02). The prevalence of MetS increased significantly from 4.58% to 9.47% in boys (*p* = 0.02). Systolic BP increased slightly from 105.20 mmHg to 106.12 mmHg (*p* = 0.34), whereas diastolic BP increased significantly from 65.91 mmHg to 68.29 mmHg in girls (*p* = 0.02). In the 10–18-year-old age group, TG values increased from 74.92 mg/dL in 2019 to 87.9 mg/dL in 2020 in girls but not in boys (*p* < 0.01 in girls, *p* = 0.23 in boys). HDL-c level decreased from 54.57 mg/dL in 2019 to 52.61 mg/dL in 2020, with a borderline significance level in girls (*p* = 0.07). The mean WC and prevalence of abdominal obesity in boys and girls aged 6–9 years, respectively, increased in 2020 compared to 2019 but not significantly.

### 3.4. Differences in Candidate Risk Factors According to Metabolic Syndrome

Analysis of data from KNHANES 2019–2020 used to compare the nutrition and lifestyle components between normal and MetS groups in youth aged 12–18 years revealed significant differences in paternal education, sports-drink intake, and proportion of carbohydrate intake (Table 4). In 2019, before COVID-19, the level of paternal education was significantly lower in the MetS group than in the normal group (*p* < 0.01). In 2020, during the COVID-19 pandemic, the level of paternal education was only slightly lower in the MetS group (*p* = 0.26). Sports- drink intake, which was investigated since 2019, was significantly higher in the MetS group (*p* < 0.01).

We conducted an analysis using odds ratios (OR) to confirm whether there were real differences in paternal education, sports-drink intake, and the proportion of carbohydrate intake. The results of the analysis showed that there was a significant association between a carbohydrate intake greater than 65% and metabolic syndrome both before and after COVID-19. Furthermore, we found a significant association between paternal education level before COVID-19 and sports-drink intake during COVID-19, as shown in Table 5.

## 4. Discussion

This is the latest study to investigate the difference in the prevalence of obesity and MetS before and after the COVID-19 pandemic outbreak using data from KNHANES 2019–2020, focusing on children and adolescents. Previous studies have examined the prevalence of obesity and MetS in pediatric populations. The overall prevalence of childhood obesity has slightly increased since the early 2000s; however, the proportion of extreme obesity has increased significantly in youth aged 10–18 years, especially among boys. The prevalence of extreme obesity in Korean youth has ranged from 3.8% to 7.7% in several studies [10,24,25]. In this study, obesity increased significantly in those aged 2–9 years, while extreme obesity increased in girls aged 2–9 years and boys aged 10–18 years. Adolescents with severe obesity are more vulnerable to metabolic syndrome than adolescents with less severe obesity [26]. Moreover, childhood obesity can lead to adulthood obesity and various cardiovascular, metabolic, and psychosocial complications [27].

When lifestyle habits and nutritional status before and after the COVID-19 pandemic were compared, significant differences were observed only in paternal education, consumption of sports drinks, and the proportion of carbohydrate intake in this data. However, there were no significant differences in other factors such as sleep time, physical activities, and sedentary time. In another study, it has been reported that sleep duration decreased by 0.05 h on average during the COVID-19 pandemic among Korean adolescents [28]. Contrary to these findings, studies in Italy, Spain, Greece, and China reported that children slept more during the COVID-19 lockdown [29]. South Korea did not enforce a nationwide “shut down” measure, except on a localized and temporary basis. For instance, in areas with high incidence rates such as Daegu, South Korea, residents were advised to stay at home and avoid leaving their residences for a minimum of two weeks [30].

The prevalence of MetS has increased in the last decade in accordance with obesity trends [31,32]. The prevalence of MetS was stable in Korean children from 1998 to 2005. The prevalence of MetS was 4.5% in 2007–2009, 3.9% in 2010–2012, 4.1% in 2013–2015, and 6.2% in 2016–2018 in KNHANES [33]. We found that the prevalence of MetS in 2020 increased to 7.8%. The prevalence of obesity showed a parallel increasing trend to that of MetS [33]. We identified a detrimental progression in MetS components, such as WC, systolic BP, diastolic BP, low-density lipoprotein-cholesterol, TGs, FBG, and HDL-c after the COVID-19 outbreak [26]. In boys, an overall increasing tendency was observed in the prevalence of MetS, with elevated WCs after the COVID-19 outbreak. In contrast, the overall prevalence of MetS did not change in girls, despite an increased mean TG level and diastolic BP.

Due to social distancing, schools were closed, and online classes led to teenagers staying home for a long time without going out. As time spent at home increased, screen time increased, and eating behaviors also changed, with increased fast food, snack, and high-calorie food intake [1]. The significant increase in obesity at a younger age after the COVID outbreak is an important public health concern, as future metabolic complications may appear earlier [7]. Moreover, younger ages of onset of obesity mean longer durations, thereby affecting MetS in adolescence [33,34]. Once obesity has developed, reverting to a normal weight status becomes challenging. Therefore, it is crucial to focus on preventing obesity from a young age. The early and accurate intervention in obesity and comorbidities is fundamental to improve health and quality of life in adults.

Unfavorable lifestyle behaviors likely lead to an increased risk of MetS in adult populations [35]. Risk factors related to MetS in this study included the degree of paternal education and the intake of soft and sports drinks and carbohydrates. Soft and sports drinks are high in calories, thereby contributing to obesity [36]. In Korea, a national survey on the frequency of beverage intake conducted in 2019 revealed a similar result [37].

There are several limitations to our study. Since this study was based on KNHANES data, nutritional details, such as vitamin, folic acid, and zinc levels, were not included. However, these details must be included in further evaluations due to their relationship with cardiovascular disease, bone mineralization, and MetS [25]. Although KNHANES is a large-scale survey, information is not collected equally for all subjects for each survey index. For example, blood tests were conducted only for those aged ≥ 10 years, and WC was measured only for those aged ≥ 6 years; thus, results at younger ages are incomplete. Thus, while KNHANES is utilized to evaluate relevance by cross-sectional examination, it is inadequate to assess causality. Although KNHANES offers representative data, the number of children and adolescents is not large. Therefore, it is necessary to confirm the results of this study through subsequent research.

The body fat percentage may be very different even if the BMI is the same, and the amount of fat mass increases the risk of MetS independently of BMI [31]. Moreover, a low muscle mass increases the risk of MetS in adolescents [38]. Therefore, we speculate that more meaningful results may be obtained if the percentage of body fat mass is studied as an indicator of obesity. In the KNHANES data, the body fat mass was measured by dual energy X-ray absorptiometry until 2011, and subsequent data were not available.

The strength of this study is that it presents the most recent and large sample number at the national level; thus, data reliability is high. In this study, representative objective data were used to investigate obesity and MetS in children and adolescents. Although many subjective surveys on weight gain have been performed since the COVID-19 outbreak, no studies in Korea have used objective indicators of BMI and MetS components. Furthermore, we evaluated the changes in the prevalence of obesity, MetS, and various health and lifestyle behaviors in children and adolescents. Preventing the progression of obesity to MetS in youth has become an urgent public health concern in Korea.

## 5. Conclusions

In conclusion, the prevalence of obesity and MetS among children and adolescents has increased after the COVID-19 outbreak and is possibly associated with reduced physical activity due to social distancing. Specifically, the increase in mean BMI and abdominal obesity among those aged 2–9 years could translate into a future health burden in Korea. Childhood obesity has significant implications for the development and progression of metabolic diseases throughout an individual’s lifespan. Therefore, it is important to pay attention to the lifestyle of obese youth and monitor childhood obesity and MetS closely.

## 6. Recommendation

Pediatricians need to devise detailed strategies to prevent and manage obesity in collaboration with experts in various healthcare fields. Early intervention and prevention strategies can reduce the long-term burden of obesity and metabolic complication. A well-balanced diet with proper protein intake should be emphasized. As vitamin D deficiency is an associated risk factor of MetS, vitamin D consumption should be encouraged [39,40].

## Figures and Tables

**Figure 1 children-10-01105-f001:**
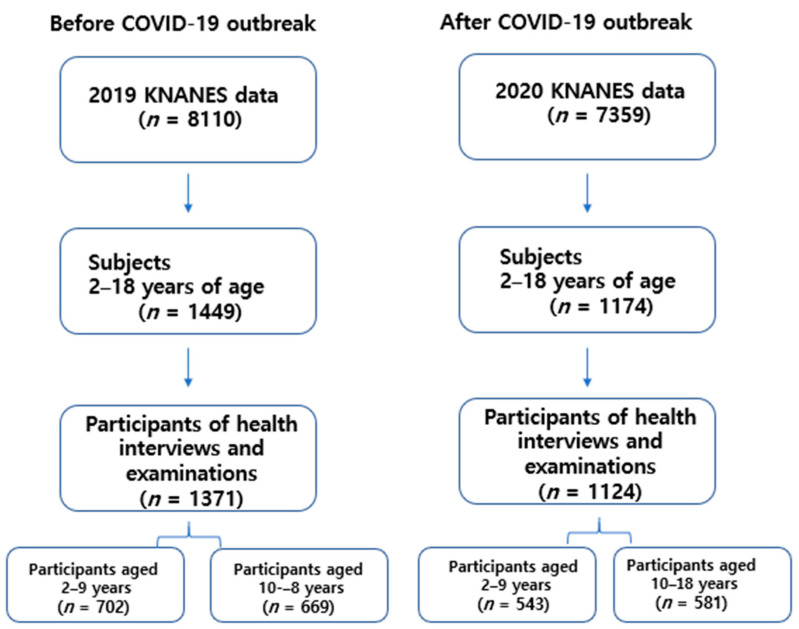
Flow Diagram for Study Participants.

**Figure 2 children-10-01105-f002:**
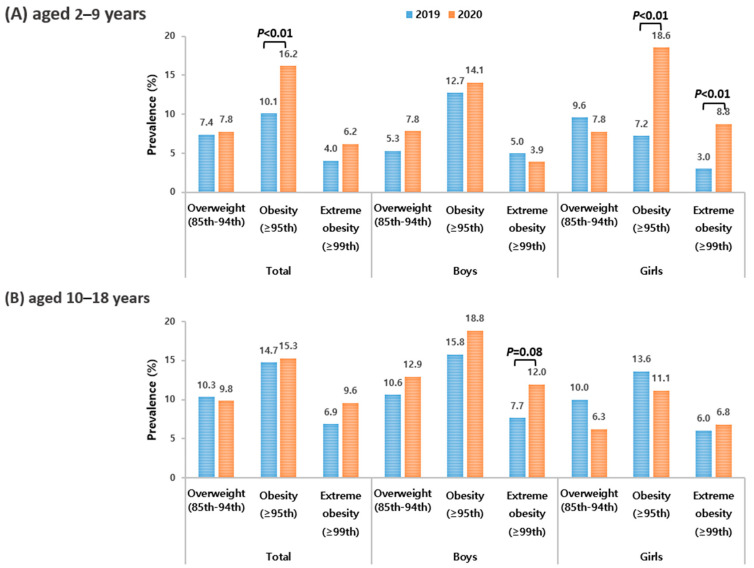
Prevalence of overweight, obesity, and extreme obesity among Korean children and adolescents aged 2–19 years before and after COVID-19. (**A**) aged 2–9 years (**B**) aged 10–18 years. Values are presented as weighted %.

**Figure 3 children-10-01105-f003:**
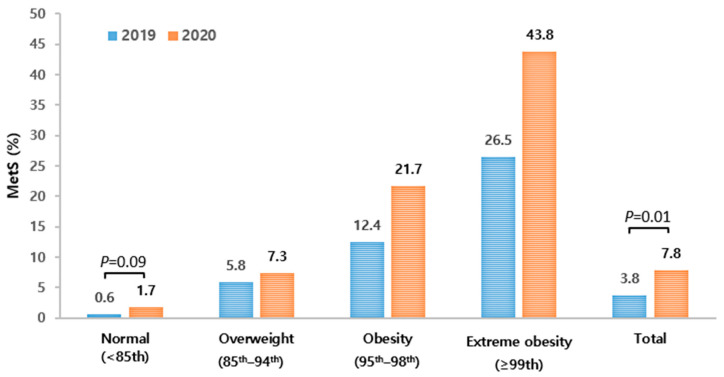
The prevalence of metabolic syndrome by BMI categories in 2019 and 2020 in youth aged 10–18 years. Values are presented as weighted %.

**Table 1 children-10-01105-t001:** Differences in basic characteristics of subjects before and after the COVID-19 outbreak by age group.

Variables	Children Aged 2–9 Years		Children Aged 10–18 Years			
	Before (2019)	After (2020)	*p*-Value	Before (2019)	After (2020)	*p*-Value
Age (years)	5.6 (5.41–5.79)	5.78 (5.56–6.01)	0.21	14.16 (13.93–14.39)	14.07 (13.82–14.32)	0.6
Sex						
Boys	357 (51.92%)	284 (52.22%)	0.92	359 (52.22%)	326 (53.41%)	0.71
Girls	345 (48.08%)	259 (47.78%)		310 (47.78%)	255 (46.59%)	
Household incomes						
Q1 (low)	48 (6.36%)	33 (5.98%)	0.54	66 (10.68%)	44 (7.14%)	0.15
Q2	224 (32.63%)	168 (29.76%)		199 (29.1%)	159 (26.88%)	
Q3	248 (35.01%)	178 (32.5%)		196 (29.61%)	217 (37.85%)	
Q4 (high)	182 (26%)	164 (31.77%)		206 (30.61%)	160 (28.12%)	
Paternal education						
Less than high school graduate	147 (29.54%)	109 (25.99%)	0.44	177 (38.76%)	159 (41.09%)	0.64
College graduate or higher	347 (70.46%)	278 (74.01%)		269 (61.24%)	212 (58.91%)	
Maternal education						
Less than high school graduate	163 (25.7%)	148 (30.32%)	0.27	253 (42.54%)	222 (43.99%)	0.75
College graduate or higher	467 (74.3%)	336 (69.68%)		350 (57.46%)	273 (56.01%)	
Body mass index (kg/m^2^)	16.53 (16.31–16.75)	17.1 (16.8–17.4)	0.00	21.25 (20.78–21.72)	21.41 (20.94–21.87)	0.64
Regular physical activity (age ≥12 years) ^a^	NA	NA	NA	14 (3.36%)	13 (2.45%)	0.54
Regular strength training (age ≥12 years) ^b^	NA	NA	NA	133 (29.12%)	137 (31.6%)	0.49
Sedentary time (h, age ≥12 years)	NA	NA	NA	11.32 (11.06–11.57)	11.19 (10.89–11.49)	0.52
Recognizing stress (age ≥12 years)	NA	NA	NA	113 (23.08%)	107 (26.21%)	0.37
Average sleep time per day of the weekday (h, age ≥12 years)	NA	NA	NA	6.87 (6.74–7)	6.82 (6.68–6.97)	0.61
Soft-drink intake ^c^						
<3 times/week	302 (82.6%)	192 (75.77%)	0.08	346 (58.09%)	252 (57.68%)	0.91
≥3 times/week	65 (17.4%)	58 (24.23%)		239 (41.91%)	190 (42.32%)	
Sports-drink intake ^c^						
<3 times/week	358 (97.58%)	234 (93.37%)	0.03	479 (81.4%)	354 (80.43%)	0.75
≥3 times/week	9 (2.42%)	16 (6.63%)		106 (18.6%)	88 (19.57%)	
Smoking (≥12 years)	NA	NA	NA	15 (2.72%)	18 (3.33%)	0.61
Nutritional factor						
Dietary supplement use	447 (66.52%)	338 (72.36%)	0.18	255 (41.77%)	200 (44.59%)	0.46
Total energy intake	1528.21 (1473.51–1582.91)	1537.11 (1476.05–1598.18)	0.84	2016.15 (1943.83–2088.46)	1910.08 (1831.89–1988.27)	**0.05**
Excess calorie intake	141 (20.71%)	102 (21.7%)	0.72	136 (24.67%)	92 (22.89%)	0.54
% of energy from carbohydrates	60.57 (59.76–61.39)	60.38 (59.44–61.31)	0.76	59.01 (58.08–59.94)	58.08 (56.98–59.18)	0.21
Carbohydrate > 65% of total energy	194 (30.34%)	135 (29.72%)	0.85	134 (22.83%)	105 (23.4%)	0.84
% of energy from fat	24.79 (24.13–25.46)	24.56 (23.72–25.4)	0.67	24.97 (24.13–25.81)	25.74 (24.86–26.62)	0.21
Fat intake > 30% of total energy	167 (23.49%)	99 (20.41%)	0.31	139 (24.01%)	117 (25.93%)	0.54

Values are presented as weighted mean (95% confidence interval) for continuous variables or unweighted frequency (weighted %) for categorical variables. *p*-values are from *t*-test for continuous variables and chi-squared test for categorical variables. ^a^ Regular physical activity was defined as physical activity for at least 60 min per day. ^b^ Regular strength training was defined as muscle training at least 2 days per week. ^c^ Soft-drink intake and sports-drink intake were measured in children aged 6 years and older.

**Table 2 children-10-01105-t002:** Comparison of metabolic syndrome components before and after the COVID-19 outbreak.

Metabolic Syndrome Components	10–18 Years		
	Before (2019)	After (2020)	*p*-Value
Waist circumference (cm)	71.9 (70.73–73.07)	72.41 (71.18–73.63)	0.56
Elevated WC (≥90th)	83 (12.59%)	97 (16.35%)	0.14
Systolic BP (mmHg) ^a^	108.71 (107.59–109.82)	109.28 (107.98–110.58)	0.51
Diastolic BP (mmHg) ^a^	66.78 (65.82–67.73)	68.56 (67.43–69.69)	0.02
Elevated BP (≥90th) ^b^	11 (1.69%)	16 (2.86%)	0.29
Fasting Blood Glucose (mg/dL)	92.46 (91.69–93.24)	92.31 (91.14–93.47)	0.83
Elevated FBG (≥100 mg/dL)	94 (13.67%)	73 (12.13%)	0.49
Triglyceride (mg/dL) ^c^	75.52 (72.08–79.12)	83.97 (79.72–88.45)	0.00
Elevated TG (≥110 mg/dL)	141 (24.2%)	152 (27.04%)	0.38
HDL-c (mg/dL)	52.62 (51.61–53.63)	51.42 (50.43–52.4)	0.01
Low HDL (<40 mg/dL)	46 (8.53%)	51 (10.19%)	0.39
cMetS score	0.03 (−0.30–0.36)	0.65 (0.29–1.01)	0.01
MetS	24 (3.79%)	38 (7.79%)	0.01

Values are presented as weighted mean (95% confidence interval) for continuous variables or unweighted frequency (weighted %) for categorical variables. ^a^ Blood pressure in the 2020 survey was measured in children aged 6 years and older. ^b^ Elevated BP was defined for children aged 7 years and older, based on the 2007 Korean Children and Adolescent National Growth Charts. ^c^ Log substitution is performed with non-normal distribution.

**Table 3 children-10-01105-t003:** Comparison of metabolic syndrome components before and after the COVID-19 outbreak by sex.

Metabolic Syndrome Components	10–18 Years					
	Boys			Girls		
	2019	2020	*p*-Value	2019	2020	*p*-Value
Waist circumference (cm, age ≥6 years)	75.03 (73.61–76.45)	76.27 (74.93–77.6)	0.22	68.47 (67.03–69.91)	67.96 (66.47–69.44)	0.63
Elevated WC (≥90th)	47 (12.52%)	63 (19.83%)	0.02	36 (12.66%)	34 (12.33%)	0.92
Systolic BP (mmHg)	111.92 (110.60–113.24)	111.83 (110.13–113.53)	0.94	105.2 (103.91–106.49)	106.12 (104.79–107.45)	0.34
Diastolic BP (mmHg)	67.57 (66.40–68.74)	68.78 (67.44–70.11)	0.19	65.91 (64.59–67.23)	68.29 (66.87–69.7)	0.02
Elevated BP (≥90th)	6 (2.01%)	12 (3.9%)	0.21	5 (1.34%)	4 (1.57%)	0.85
Fasting Blood Glucose (mg/dL)	92.84	92.96	0.88	92.05	91.56	0.68
	(91.87–93.82)	(91.91–94.02)		(90.92–93.19)	(89.62–93.51)	
Elevated FBG (≥100 mg/dL)	51 (13.87%)	50 (15.19%)	0.68	43 (13.46%)	23 (8.68%)	0.08
Triglyceride (mg/dL) ^c^	76.08 (71.62–80.81)	80.63 (75.05–86.63)	0.23	74.92 (70.29–79.85)	87.9 (82.88–93.23)	0.00
Elevated TG (≥110 mg/dL)	78 (25.10%)	83 (27.2%)	0.63	63 (23.25%)	69 (26.87%)	0.39
HDL-c (mg/dL)	50.8 (49.60–52.00)	50.35 (49.11–51.6)	0.62	54.57 (53.01–56.13)	52.61 (51.16–54.07)	0.07
Low HDL-c (<40 mg/dL)	32 (11.58%)	35 (14.13%)	0.41	14 (5.27%)	16 (5.75%)	0.83
MetS	17 (4.58%)	25 (9.47%)	0.02	7 (2.96%)	13 (5.74%)	0.2

Values are presented as weighted mean (95% confidence interval) for continuous variables or unweighted frequency (weighted %) for categorical variables. ^c^ Log substitution is performed with non-normal distribution.

**Table 4 children-10-01105-t004:** Differences in dietary nutrition and lifestyle components according to metabolic syndrome in Korean teenagers (12–18 years of age).

	Before COVID-19 Outbreak (2019)			After COVID-19 Outbreak (2020)		
	Normal	MetS	*p* Value	Normal	MetS	*p* Value
Age	15.27	15.2	0.87	15.12	15.11	0.98
	(15.1–15.45)	(14.25–16.14)		(14.93–15.32)	(14.21–16.01)	
Boys	226 (52.27%)	13 (61.57%)	0.45	201(54.06%)	22 (67.59%)	0.15
Household incomes						
Q1 (low)	39 (10.6%)	3 (19.43%)	0.6	23 (5.76%)	3 (9.28%)	0.83
Q2	127 (28.08%)	4 (30.3%)		99 (27.78%)	6 (21.47%)	
Q3	128 (30.19%)	7 (32.76%)		133(38.65%)	16 (40.46%)	
Q4 (high)	141 (31.12%)	5 (17.51%)		98 (27.81%)	8 (28.8%)	
Paternal education						
Less than high school graduate	111 (36.74%)	10 (95.97%)	0.00	105(42.83%)	12 (57.33%)	0.26
College graduate or higher	176 (63.26%)	1 (4.03%)		133(57.17%)	7 (42.67%)	
Maternal education						
Less than high school graduate	176 (43.92%)	9 (58.4%)	0.38	149(45.97%)	13 (48.37%)	0.84
College graduate or higher	220 (56.08%)	6 (41.6%)		152(54.03%)	14 (51.63%)	
Regular physical activity	12 (3.33%)	1 (3.34%)	0.99	12 (2.7%)	0 (0.00%)	NA
(age ≥12 years)						
Regular strength training	120 (30.08%)	6 (25.11%)	0.63	112 (30.7%)	10 (30.69%)	0.99
(age ≥12 years)						
Sedentary time	11.3	11.73	0.42	11.25	10.81	0.34
(hours, age ≥12 years)	(11.03–11.57)	(10.68–12.78)		(10.9–11.6)	(9.96–11.67)	
Recognizing stress	96 (21.47%)	6 (31.79%)	0.29	90 (26.54%)	5 (14.14%)	0.15
(age ≥12 years)						
Average sleep time per day			0.72	6.75	7.02	0.32
(h, age ≥12 years)	6.85	7.04		(6.58–6.92)	(6.52–7.53)	
	6.71–6.99)	(6.03–8.06)				
Soft-drink intake						
<3 times/week	210 (56.36%)	4 (30%)	0.07	144 (53.72%)	11 (51.22%)	0.82
≥3 times/week	162 (43.64%)	13 (70%)		131 (46.28%)	10 (48.78%)	
Sports-drink intake						
<3 times/week	296 (78.71%)	14 (86.05%)	0.55	208 (77.33%)	12 (53.57%)	0.01
≥3 times/week	76 (21.29%)	3 (13.95%)		67 (22.67%)	9 (46.44%)	
Smoking	15 (3.95%)	0 (0.0%)	NA	16 (4.82%)	0.00%	NA
Nutritional factor						
Dietary supplement use	151 (39.49%)	2 (13.55%)	0.05	101 (37.52%)	10 (46.77%)	0.42
Total energy intake	2037.5 (1942.68–2132.32)	2169.4 (1835.98–2502.81)	0.44	1941.36 (1842.46–2040.26)	2131.52 (1762.94–2500.11)	0.32
Excess calorie intake	88 (24.66%)	6 (42.08%)	0.18	62 (24.71%)	5 (27.6%)	0.78
% of energy from	58.69	60.16	0.53	58.21	61.31	0.24
carbohydrates	(57.57–59.8)	(55.64–64.67)		(56.85–59.57)	(56.21–66.42)	
Carbohydrate >65% of total energy	79 (20.64%)	7 (47.61%)	0.03	75 (25.83%)	7 (38.53%)	0.25
% of energy from fat	25.21	23.73	0.55	25.73	22	0.05
	(24.2–26.21)	(19–28.45)		(24.56–26.9)	(18.22–25.77)	
Fat intake > 30% of total energy	89 (24.12%)	5 (25.31%)	0.92	75 (26.19%)	5 (16.57%)	0.32

Values are presented as weighted mean (95% confidence interval) for continuous variables or unweighted frequency (weighted %) for categorical variables.

**Table 5 children-10-01105-t005:** Assessment of risk factors for metabolic syndrome in Korean teenagers (12–18 Years age).

	Before COVID-19 Outbreak (2019)				After COVID-19 Outbreak (2020)			
	OR (95% CI)	*p*-Value	AOR (95% CI)	*p*-Value	OR (95% CI)	*p*-Value	AOR (95% CI)	*p*-Value
Girls	0.68 (0.26–1.83)	0.45	0.45 (0.07–2.77)	0.39	0.56 (0.26–1.24)	0.15	0.82 (0.18–3.86)	0.80
Age	0.98 (0.78–1.24)	0.87	1.23 (0.71–2.13)	0.46	1 (0.8–1.25)	0.98	1.21 (0.8–1.82)	0.37
Total energy intake (per 100 kcal)	1.02 (0.97–1.07)	0.43	1.02 (0.96–1.09)	0.46	1.03 (0.98–1.09)	0.29	1.02 (0.95–1.1)	0.52
Paternal education ≥College graduate or higher	0.02 (0–0.22)	0.00	0.03 (0–0.27)	0.00	0.56 (0.2–1.55)	0.26	0.54 (0.17–1.78)	0.31
Soft-drink intake ≥3 times/week	3.01 (0.85–10.71)	0.09	1.73 (0.33–9.1)	0.52	1.11 (0.46–2.64)	0.82	0.48 (0.17–1.39)	0.18
Carbohydrate >65 % of total energy	3.5 (1.11–11.01)	0.03	7.04 (1.29–38.43)	0.02	1.8 (0.65–4.95)	0.26	3.81 (1.03–14.11)	0.04
Dietary Supplement (yes vs. no)	0.24 (0.05–1.15)	0.07	0.38 (0.04–3.65)	0.40	1.46 (0.58–3.7)	0.42	2.17 (0.64–7.28)	0.21
Sports-drink intake ≥3 times/week	0.6 (0.11–3.22)	0.55	0.31 (0.03–3.13)	0.32	2.96 (1.25–7)	0.01	5.9 (1.4–24.87)	0.02

Adjusted odds ratios were estimated considering all variables presented in the table: sex, age, total energy intake, paternal education level, soft-drink intake, carbohydrate intake, dietary supplement use, and sports-drink intake.

## Data Availability

Data supporting reported results can be found via publicly available datasets (https://knhanes.kdca.go.kr/knhanes/sub03/sub03_02_05.do (accessed on 14 February 2022).

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
