# Peer review of "Increase of Prevalence of Obesity and Metabolic Syndrome in Children and Adolescents in Korea during the COVID-19 Pandemic: A Cross-Sectional Study Using the KNHANES"

_children, 2023, doi:10.3390/children10071105_

Round 1

Reviewer 1 Report

The manuscript is dealing with an interesting topic dealing with investigating the correlation between COVID-19 and prevalence of obesity and metabolic syndrome of children and adolescents in Korea. The manuscript requires certain modifications to be performed through the following raised comments;  

1-Page 2 line 81 use Body mass index instead of BMI at the beginning of the sentence

2- Page 2 line 81; “The measured body weight (kg) by the square root 81 of the measured height (m2)” this is a wrong method of calculation which raised a concern about the values presented in the manuscript.it is well known that BMI is calculated by “weight in kilograms divided by height in meters squared”.

3-Medical history and medication history were not collected or mentioned in the study which were considered very important items affecting study interpretation. Please clarify why these important information was not collected in the survey?

4-Study flow chart was not present in the manuscript which is better to be submitted.

5-Sample size calculation was not performed. The authors SHOULD clarify on what basics they had chosen the number of patients.

6-Inclusion and exclusion criteria were not mentioned in the study. This raised a concern about the wide variabilities between the subjects included in the study, please justify this point with references.

7-The use of supplementation as well as social habits including smoking were ignored in the study which might be cause of weight gain as well as weight loss.

8- Page 6 , Line 144; the authors mentioned that “There was no significant difference in physical activity, stress, and sleep time as lifestyle behaviors before and after the COVID-19 outbreak.” which sounds strange during pandemic especially there were a long SHUT DOWN periods which participate in the large change in all theses properties EXCEPT in case there was no SHUT DOWN periods applies at Korea and this SHOULD be clarified in the manuscript.

9- Discussion section did not include all the results obtained in the study. Interpretation to all the obtained results are required to be submitted.

10-Lines 303 to 309 are not correlated to the conclusion and SHOULD be removed from the conclusion section.

11-Lines 318 to 320 SHOULD be removed from conclusion section and transferred under recommendation section.  

12- The following paragraph has to be rewritten in the recommendation section;

13--The effect of healthy diet as well as antioxidants are an important item to be mentioned in the introduction as follows;

Healthy diet including fruits and vegetables, can supply the body with beneficial nutrients and antioxidants [1] including coenzyme Q10 and alpha-tocopherol which  proved to have a protective effects of antioxidants [2]. Finally, vitamin D is one of essential vitamins was deficient in children but also contribute to pathogenesis of different diseases in children [3]. Moreover, massive use of antibiotics was proven to affect gut microbiota abundance and balance leading to several inflammatory diseases as well as bacterial and viral infections [4].

References

[1] Sara AR, Mohamed Raslan, Eslam M Shehata and Nagwa A Sabri. Impact of Applied Protective Measures of COVID-19 on Public Health. Acta Scientific Pharmaceutical Sciences 5.7 (2021):63-72.

14- Sex effect as well as genetic factors which causes increasing incidence of COVID 19 was missed; the following data are recommended to be added in the introduction;

It was found that both Angiotensin Converting Enzyme 2 (ACE2) and Trans-membrane protease serine type 2 (TMPRSS2) over gene-expression in males, may raise the hypothesis of male predominance in pandemic disease, along with the higher severity and worse outcome (5). As both ACE2 and TMPRSS2 are expressed in the testes and in the prostate, thus, zinc supplementation for males will provide an additional benefit besides its antiviral effect by protecting the male reproductive system from the possible viral attack. (6)

References

DOI: 10.1016/j.retram.2020.08.002

15-The authors did not mention anything concerning the different modalities used for treatment of COVID-19 in pediatrics. A paragraph concerning this information is required to be added in the introduction very briefly.

The quality of English language is fine 

Reviewer 2 Report

In this study, they investigated the prevalence of obesity and MetS before and during COVID-19 pandemic outbreak using representative data on youth aged 2‒18 years from KNHANES conducted in 2019‒2020. Their findings are interesting, since during the COVID-19 pandemic there were changes in behavior and eating habits in the population, including the child population. However, there are several observations that need to be reviewed. Here are my comments

Major Comments:

Introduction. The authors should explain the non-pharmacological interventions used during the COVID-19 pandemic in 2020 such as isolation and other restrictions imposed by their government, which led to changes in behavior and eating habits.

Methods. Design. Include study design. Is it a retrospective or prospective cohort, or is it a cross-sectional study in two time periods? This has to be explained in detail.

Verify if they comply with all the items on the STROBE checklist.

Statistical analysis. Include the 95% CI.

Results

Lines 140-141. The authors explain that "The mean BMI increased significantly among youth aged 2‒9 years from 16.53 kg/m2 in 2019 and 17.1 kg/m2 in 2020 (p<0.01)".

However, the authors do not explain the study design. This statement (Lines 140-141) would be correct in a cohort study. On the contrary, if it is a cross-sectional study, the result does not show an increase, but rather that body weight was significantly higher during the COVID-19 pandemic compared to 2019.

Review the same statement in lines 146-147 "the overall energy intake at the age of 10‒18 years decreased significantly from 2016.15 kcal in 2019 to 1910.08 kcal in 202".

The authors should also compare the results of Figure 1. Check if they are increases in the rates of a cohort or if rather it is that the prevalence was higher in 2020, compared to 2019.

Comparisons of numerical variables should be expressed as mean plus standard deviation in Tables 1 and 2. The authors only include means and ranges.

Table 2 and Figure 2. Obviously, the MetS rate was higher in 2020, however, it is necessary to determine if it is the same cohort or the values are from different populations.

Table 4. Although there were differences in the paternal less than high school graduate, Soft drink intake, Sports drink intake, and Carbohydrate >65% of total, these values have only been measured with the Chi-square or Fisher test, the authors should perform crude OR and adjusted OR analyzes with some confounding factors to test whether these differences are real.

Discussion

Authors must include a paragraph explaining the implications of their findings for public health in Korea.

They should also discuss the strengths of their findings.

They should also revise a few paragraphs based on the comments in the Results section.

Conclusion

Lines 303-309. The conclusion should focus on your findings, not discussion paragraphs.

Authors should restate their conclusions on the basis of previous comments.

Minor comments:

Include the study design in the title.

Introduction. Lines 37-42. These sentences need to be referenced with previous studies to confirm this.

Include a stronger justification and explain what the findings of your study would serve.

Reviewer 3 Report

This paper is a very valuable contribution to the relationship between the COVID-19 pandemic and childhood obesity.  The weakest part of the paper is the introduction - this needs assistance with the  English language primarily with sentence structure.

Page 1 line 33 - add China after Wuhan

line 36 add due to the need for social distancing

line 36 - it's not clear with the "first online school". Do the authors mean online education began for the country or the first school in the country - just clarify pleas

line 38-40 - need to restructure these sentences

Page 2: line 47 - this sentence needs a reference

line 53 - ofter obesity needs a reference

line 79 - add respectively at the end

page 3: lines 101-105 seems out of place

lines 117-118-can this be better clarified?

Page 4 Table 1

Note that age is in years, provide income levels for Q1 etc 

Page 10line 269 - change wording of "It is not easy to go back after obesity has occurred". change to a more professional wording

Line 273 instead of poor use unhealthy?

Page 11 lines 310-311 - something is missing

Minor issues in introduction

Round 2

Reviewer 1 Report

The quality of the manuscript was greatly improved and all the required modifications were performed, thanks and Good Luck!

Author Response

Thank you for your comments

Reviewer 2 Report

The authors have reviewed the manuscript, but some corrections have not been included in the manuscript:

Write a more attractive and focused title, for example: Increase of prevalence of obesity and metabolic syndrome in children and adolescents in Korea: A Cross-sectional study using the KNHANES during the COVID-19 pandemic

They have included the STROBE Checklist in the response letter, but not in the manuscript or supplementary material.

Statistic analysis. Explain in the adjusted analysis (AOR) which were the adjustment variables "Adjusted odds ratios were estimated considering all variables presented in the table".

Table 5, has not been included in the manuscript.

There are several grammatical errors.
